# Successful Full-Term Delivery via Selective Ectopic Embryo Reduction Accompanied by Uterine Cerclage in a Heterotopic Cesarean Scar Pregnancy: A Case Report and Literature Review

**DOI:** 10.3390/diagnostics12030762

**Published:** 2022-03-21

**Authors:** Hyoeun Kim, Ji Hye Koh, Jihee Lee, Yeongeun Sim, Sang-Hun Lee, Soo-Jeong Lee, Jun-Woo Ahn, Hyun-Jin Roh, Jeong Sook Kim

**Affiliations:** Department of Obstetrics and Gynecology, University of Ulsan College of Medicine, Ulsan University Hospital, Ulsan 44033, Korea; ogkimhe@gmail.com (H.K.); 0735445@uuh.ulsan.kr (J.H.K.); byjhsy@gmail.com (J.L.); yeongeun.sim@gmail.com (Y.S.); shlee73@uuh.ulsan.kr (S.-H.L.); exsjlee@uuh.ulsan.kr (S.-J.L.); ahnjwoo@uuh.ulsan.kr (J.-W.A.); 0729345@uuh.ulsan.kr (H.-J.R.)

**Keywords:** heterotopic cesarean section scar pregnancy, ectopic pregnancy, selective embryo reduction, arteriovenous malformation

## Abstract

Heterotopic cesarean scar pregnancy (HCSP) is a combination of cesarean scar pregnancy (CSP) and intrauterine pregnancy (IUP). Cesarean scar pregnancy is accompanied by life-threatening complications, such as uterine rupture and massive bleeding. Herein, we present a case of HCSP treated with selective potassium chloride injection into the CSP under ultrasonography in association with uterine cerclage to control vaginal bleeding; this led to a successful IUP preservation and full-term delivery. Additionally, we will review several previous reports on HCSP management, including our case.

## 1. Introduction

Heterotopic cesarean scar pregnancy (HCSP) is a rare cesarean scar pregnancy (CSP) combined with an intrauterine pregnancy (IUP), and is accompanied by life-threatening complications, such as uterine rupture and massive bleeding [1,2]. The incidence of HCSP in natural pregnancy is extremely low; however, the number of cesarean sections and the expansion of assisted reproductive technology (ART) have gradually increased [3]. Nevertheless, the preservation of concurrent IUP and fertility remains a challenge because of the absence of a standard protocol for HCSP management [2,3]. Herein, we present a case of HCSP that was treated to preserve the IUP and the patient’s fertility. In the first trimester, a selective ectopic embryo reduction in the CSP was performed using intrathoracic potassium chloride (KCl) injection and embryo aspiration. In the second trimester, the remaining gestational tissue growth in the CSP, as well as the occurrence of vaginal bleeding, was controlled via a uterine cervical cerclage. Finally, full-term delivery was successfully achieved without uterine arterial embolization or hysterectomy by repeated cesarean sections.

## 2. Case Report

A 36-year-old woman (gravida 2, para 1) was transferred from a local hospital because of a cesarean ectopic pregnancy with IUP. The patient underwent in vitro fertilization-embryo transfer (IVF-ET) using two ova. Two years ago, she delivered a baby via a lower segment cesarean section. Ultrasonography at 6^+1^ gestational weeks (GW) revealed two gestational sacs; one in the uterine fundus and the other in the anterior uterine isthmus (Figure 1a,b). Both had fetal cardiac activity, and the mother had no vaginal bleeding or abdominal pain.

The patient decided to undergo selective embryo CSP reduction to preserve the normal fetus. Under spinal anesthesia, a uterine sound was inserted to reach the CSP (the anterior uterine isthmus) in the uterus at 6^+3^ GW under ultrasonographic guidance (Figure 2a). A 20 cm long 18-gauge spinal needle, which was bent at an equal angle to the sound, was then guided along to reach the CSP (Figure 2b). About 0.1 mL of 2 mEq/mL KCl was slowly loaded via the bent 18-gauge spinal needle into the fetal heart of the CSP (Figure 2c). After cardiac arrest of the CSP, the expired embryo was completely aspirated without affecting the placenta around the CSP. No immediate complications were observed in either the mother or the normal IUP after treatment.

The following day, sonography showed an absence of cardiac activity in the CSP, whereas the IUP was alive. Five days after the procedure, a 4 × 1.7 cm placenta with hematoma was detected at the reduction site using sonography, while the normal fetus in the uterine fundus was stable (Figure 3).

A sonographic examination at 10 GW revealed that the placenta around the reduction site had grown into the uterine cervix, resulting in intermittent vaginal spotting and a shortening of the cervical length. Considering a cervical length of less than 10 mm and the protrusion of the remnant placenta into the internal os (Figure 4a), a uterine cervical cerclage was inserted at 12 GW by placing the cervical intruding placenta inside the uterine cavity. The purpose of the cerclage was to control the growth of the remaining placenta in the CSP, as well as vaginal bleeding (Figure 4b,c). McDonald operation with double ligations using braided polyester thread (Ethibond^TM^, Ethicon, New Jersey were implemented in the cervical cerclage [4]. A previous retrospective study reported that the braided thread suture in the cervical cerclage showed an improvement in neonatal survival, the prevention of preterm birth before 28 GW, less PPROM, and maternal febrile morbidity, compared to Mersilene tape (Mersilene™, Ethicon, Somerville, NJ, USA). [5]. Ultrasonography at 24^+3^ GW showed dilated and tortuous blood vessels encompassing the lesion (10 × 6 × 3 cm^3^ in volume), suggestive of an enlarged arteriovenous malformation (AVM) (Figure 4d). During the antenatal period, there were no severe complications, including preterm labor and short cervical length.

The delivery was performed at another hospital for private reasons. A healthy female baby weighing 2415 g was delivered via elective cesarean section at 37^+6^ GW. After the delivery of the baby, massive bleeding developed at the site of the CSP. The RGT was removed, and the bleeding focus was controlled by multiple sutures. Her vital signs were stable with a red blood cell transfusion. She did not need further intensive care. The RGT was histologically confirmed as an AVM.

## 3. Discussion

Heterotopic cesarean scar pregnancy is one of the rarest heterotopic pregnancies, and it requires the careful management of a viable IUP [3,6,7]. The incidence of heterotopic pregnancy is estimated to be 1 in 30,000 deliveries. However, ART has increased the incidence of heterotopic pregnancies to 1% [3,8]. Attempts have been made to identify the cause of CSP at a molecular level. For example, a previous study showed that the endometrial expression of the integrin β3 subunit and leukemia inhibitory factor (LIF) was positively correlated with endometrial receptivity and embryo implantation [9]. In particular, their expression in the cesarean scar was significantly higher than in the endometrium of the uterine cavity. Heterotopic cesarean scar pregnancy poses a higher risk of antenatal events, such as vaginal bleeding, fetal demise of the IUP, and uterine rupture [1,7,10,11,12].

Our comprehensive literature survey of 46 HCSP cases published in journals confirmed that 32 patients reported HCSP related to ART (Table 1) [1,2,3,6,7,10,12,13,14,15,16,17,18,19,20,21,22,23,24,25,26,27,28,29,30,31]. Advances in ultrasonography have facilitated the early detection of HCSP, even during the embryonic period [6,22]. Although a standard protocol for managing HSCP remains inconclusive, most procedures focus on the selective reduction of the ectopic CSP [2,31]. The treatment modality for selective embryo/fetal reduction usually involves either an ultrasound (US)-guided intervention or surgical intervention (or both). Among the 42 cases, from which four artificial abortions were excluded, 19 (45%) cases were expectant management and 16 (38%) cases were US-guided interventions (Table 1). Surgical intervention accounted for only seven cases (17%). The US-guided interventions included the injection of embryocidal drugs, gestational sac (GS) aspiration, or a combined procedure. The US-guided interventions were performed between 8^+2^ and 10^+2^ GW (median: 8^+4^ GW). Potassium chloride was used in all injections, except for one case where methotrexate (MTX), which can cause teratogenicity to the normal IUP, was co-administered [18]. The KCl injection facilitated the spontaneous regression/detachment of the demised CSP; however, 12 cases among the US-guided interventions reported that the remnant gestational tissue (RGT) still existed. The persistence of RGT in CSP can lead to various complications, such as vaginal bleeding, probably due to its vascular characteristics [1,2,7,13,14,15,18,20,21,22,25]. Indeed, seven cases reported some problematic concerns (Table 1). For example, four cases documented vaginal bleeding after the intervention. Three cases described the development of RGT into AVM, among which one case was accompanied by a morbidly adherent placenta (MAP) that eventually required a forced hysterectomy [25]. The other RGT-to-AVM case involved a friable vascular mass with dilated vessels in a repeat cesarean section, in which uterine artery embolization was performed to control bleeding [22].

As described before, US-guided intervention can lead to RGT persistence, which can develop into AVM, accompanied by various complications, including vaginal bleeding, weakness of the scar site, or incomplete scar rupture. Sonography-guided selective CSP embryo reduction, which was successfully employed in our case, has been recommended for HCSP management because of its easy manipulation, high IUP success rate, and fewer complications, although it cannot exclude RGT persistence [33]. We implemented cervical cerclage after a selective CSP embryo reduction to prevent cervical shortening caused by AVM; this eventually controlled vaginal bleeding and promoted a successful full-term delivery. It can be speculated that surgical cerclage may tighten the loosened cervical canal, counteract the outward pressure formed by AVM, and preclude massive hemorrhage during the pregnancy (Figure 2c). Although a standard procedure has not been established, the first application of cervical cerclage in HCSP management in our case was helpful in managing fertility and IUP survival, as well as in reducing complications.

The fundamental surgical intervention for ectopic CSP is the direct excision of the ectopic mass at the cesarean scar via an open laparotomy, hysteroscopy, laparoscopy, or dilatation and curettage. Our literature review identified seven cases of surgical intervention. The surgical removal of CSP is a feasible way to prevent antenatal complications, such as vaginal bleeding and RGT growth [10,30]. Moreover, pelviscopic excision can reinforce the lower uterine segment [30]. An open laparotomy would provide more security to the previous scar because the extent of the operation field becomes broader and the operator can handle profuse bleeding more easily [21]. It is of note that the surgical approach for RGT removal should be considered cautiously in terms of preserving fertility. One patient experienced uterine rupture, and the normal fetus ended with early preterm birth before 24 weeks, even after surgical repair [5]. Another case involved laparoscopic excision after a US-guided intervention, owing to the growth of an ectopic mass and MAP-like sonographic findings. However, the surgery resulted in a hysterectomy because of uncontrolled bleeding [23].

Another option for HCSP was expectant management (19 cases) [4,18,25,28]. Nine cases confirmed the absence of a heartbeat in their CSPs, whereas seven cases reported live births and two cases did not. Among the two deaths, one was an induced abortion due to the premature preterm rupture of the membrane in early IUP, and the other was a uterine rupture accompanied by severe bleeding at 12 GW. The latter case involved a laparotomy for repair, but the IUP did not survive [28]. Among the 10 cases with a heartbeat in their CSPs, three reported miscarriages in the CSP but live IUPs were delivered [28]. Additionally, one case documented massive bleeding at eight GW during expectant management and underwent hysteroscopic excision [28]. Two vital babies from CSPs were successfully delivered in two cases at 37 and 40 GW [28]. However, severe postpartum bleeding due to placenta accreta occurred in one case, which was managed by the excision of the uterine anterior lower segment and uterine artery ligation. The other patient also had focal placenta accreta [28]. One case reported the implementation of expectant management for a successful delivery of twins [4]. Nevertheless, a recent study on the association between poor obstetric outcomes and HCSP demonstrated that the gestational age at treatment and a higher number of previous CSs were related to antepartum and postpartum hemorrhage, irrespective of the treatment mode [29].

## 4. Conclusions

As described before, US-guided intervention can lead to RGT persistence, which can develop into AVM, accompanied by various complications, including vaginal bleeding, weakness of the scar site, or incomplete scar rupture. Sonography-guided selective CSP embryo reduction, which was successfully employed in our case, has been recommended for HCSP management because of its easy manipulation, high IUP success rate, and fewer complications, although it cannot exclude RGT persistence [30]. Herein, we implemented a cervical cerclage after a selective CSP embryo reduction to prevent cervical shortening caused by AVM; this eventually controlled vaginal bleeding and promoted a successful full-term delivery. It can be speculated that surgical cerclage may tighten the loosened cervical canal, counteract the outward pressure formed by AVM, and preclude massive hemorrhage during the pregnancy (Figure 2c). Although a standard procedure has not been established, the first application of cervical cerclage in HCSP management in our case was helpful in managing fertility and IUP survival, as well as in reducing complications.

## Figures and Tables

**Figure 1 diagnostics-12-00762-f001:**
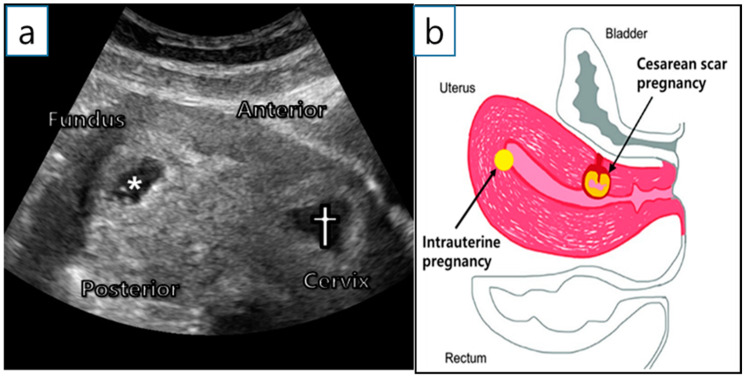
Initial examination. (**a**) Initial transvaginal ultrasound examination at 6^+1^ GW. *—intrauterine gestational sac; †—CSP. (**b**) Description of sagittal plane.

**Figure 2 diagnostics-12-00762-f002:**
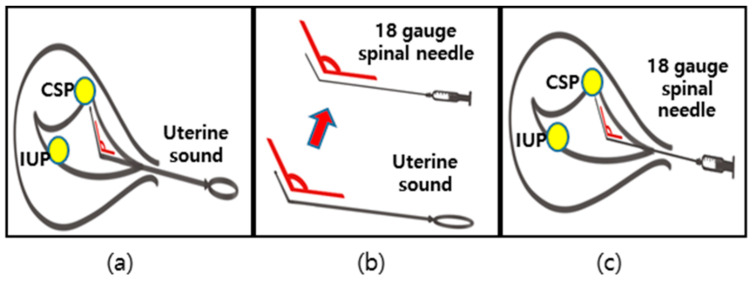
Schematic procedures for embryo reduction. (**a**) A uterine sound was inserted to reach the CSP. (**b**) A spinal needle bent at an equal angle to the sound was then guided along to locate the CSP. (**c**) Potassium chloride was injected via spinal needle into the CSP.

**Figure 3 diagnostics-12-00762-f003:**
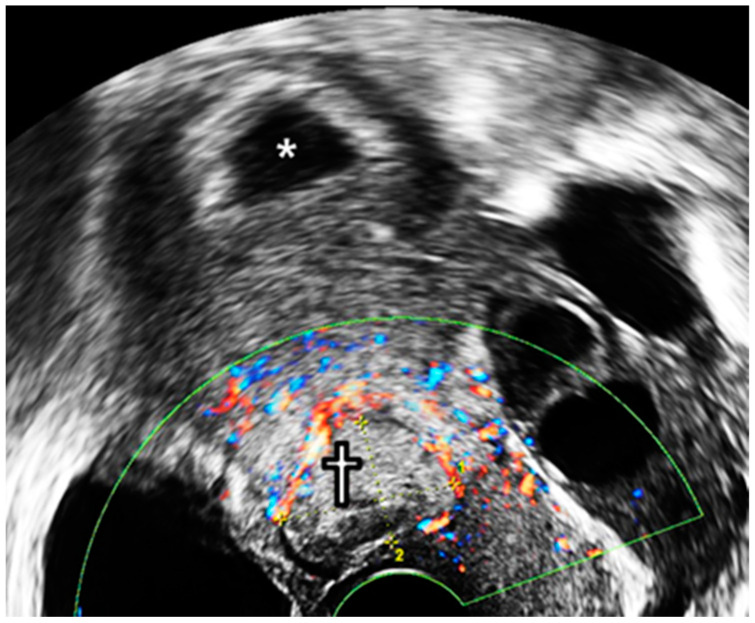
Transvaginal ultrasonography examination at 7^+1^ GW. *—intrauterine gestational sac; †—RGT; CSP—cesarean scar pregnancy; GW—gestational weeks; RGT—remnant gestational tissue.

**Figure 4 diagnostics-12-00762-f004:**
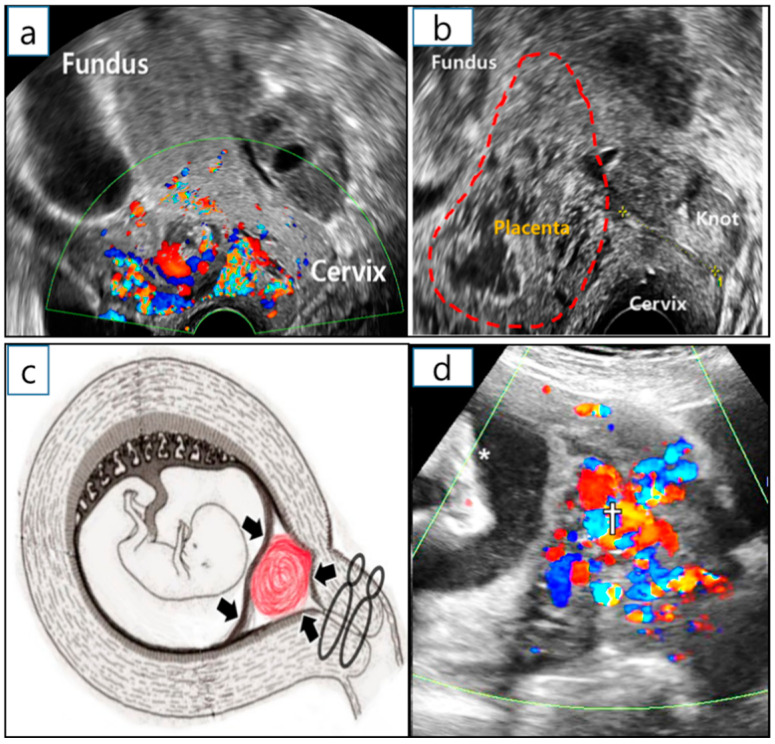
Management of arterio-venous malformation after selective CSP embryo reduction. (**a**) Ultrasonography to visualize RGT with cervical shortening at 10^+3^ GW before cervical cerclage. The RGT went into the uterine cervical internal os. (**b**) Transvaginal ultrasonography at 12^+3^ GW after the cerclage. (**c**) Effect of cervical cerclage role on HCSP management (details are written in the discussion section). (**d**) Ultrasound examination at 24^+3^ GW. Enlarged arterio-venous circulation was observed in the demised CSP (†); asterisk (*) indicates the fetal foot. CSP—cesarean scar pregnancy; GW—gestational weeks; RGT—remnant gestational tissue.

**Table 1 diagnostics-12-00762-t001:** Literature review of heterotopic cesarean scar pregnancy.

Reference	Conception/Previous CS (n)	Diagnosis Modality/GW/Symptoms or Event	Cardiac Activity of CSP/IUP	Management/GW	RGT	Antenatal Event	Pregnancy Outcome
Salomon [13], 2003	ART/1	TVUS/8/None	Yes/Yes	US-guided intervention (KCL injection)/NM	Persistent	PROM	CS at 36 GW, live female, 2800 g, RGT excision during CS
Yazicioglu [14], 2004	Spontaneous/1	TVUS/6^+2^/VB	Yes/Yes	US-guided intervention (KCL injection)/7^+2^	Spontaneously disappeared	PROM	CS at 30 GW, live male, 1530 g, RGT detachment without complications
Hsieh [10], 2004	ART (twin IUPs + CSP)/2	TVUS/6/VB	Yes/Yes	US-guided intervention (EA)/NM	Spontaneously disappeared	Preterm labor	CS at 32 GW
Wang [1], 2007	ART/3	TVUS/7/None	Yes/Yes	US-guided intervention (KCL injection)/NM	Persistent	Preterm labor	CS at 35 GW, live male, 1820 g, bilateral internal iliac artery ligation due uterine bleeding, RGT excision during CS
Demirel [12], 2009	N/M/1	TVUS/6^+5^/VB	Yes/Yes	Surgical intervention (laparoscopy)/NM	Removed	None	CS at 38 GW, live singleton
Taşkin [15], 2009	N/M/1	TVUS/8^+4^/VB	Yes/Yes	US-guided intervention (KCL injection)/9	Persistent	Preterm labor	CS at 34 GW, live female, 2310 g, RGT excision during CS
Wang [16], 2010	ART/1	TVUS/7/VB	Yes/Yes	Surgical intervention (hysteroscopy)/7	Removed	None	CS at 39 GW, live male, 3250g
Gupta [17], 2010	ART/4	TVUS/6^+1^/None	Yes/Yes	US-guided intervention (EA)/6^+3^	Persistent	None	Termination at 12 GW due to trisomy 13
Litwicka [18], 2011	ART/1	TVUS/6/None	Yes/Yes	US-guided intervention (KCl + MTX injection)/8	Persistent	Placental abruption	CS at 36 GW, 1990 g male, Miller syndrome
Dueñas-Garcia and Young [19], 2011	Spontaneous/3	TVUS, MRI/5/None	Yes/Yes	MTX + leucovorin (used for abortion)/NM	NM		
Ugurlucan [3], 2012	ART/1	TVUS/6/VB	Yes/Yes	US-guided intervention (KCl injection + EA)/NM	None	None	CS at 38 GW, live singleton, subtotal hysterectomy due to postpartum bleeding
Bai [20], 2012	ART/1	TVUS/7^+6^/VB	Yes/Yes	Expectant	Persistent	CSP miscarriageat 8^+4^ GW, VB and protruding RGT	CS at 36^+4^ GW due to preterm labor, live male, 2950 g
Uysal [21], 2013	Spontaneous/2	TVUS/8/None	Yes/Yes	US-guided intervention (KCL injection)/NM	Persistent	Preterm labor	CS at 35 GW, live female, 2480 g, incomplete uterine rupture, RGT excision during CS
Lui [22], 2014	ART/1	TVUS/5/VB	Yes/Yes	US-guided intervention (repeated EA)/NM	Persistent	None	CS at 37 GW, live female, 2660 g, RGT with AVM, selective UAE
Kim [6], 2014	Spontaneous/2	TVUS/5^+5^/None	Yes/Yes	Expectant	Persistent	None	CS at 37^+2^ GW, live twins, bladder adhesion, placenta accreta, bilateral uterine artery ligation
Armbrust [23], 2015	ART/2	TVUS/7/None	Yes/Yes	Surgical intervention (laparotomy)/NM	None	None	CS at 37^+2^ GW, live singleton, 2895 g
Yu [2], 2016	ART/1	TVUS/11/None	Yes/Yes	US-guided intervention (KCl)/16^+4^	Persistent	PPT, accreta	CS at 37^+6^ GW, live male, 2890 g, profuse vascularization with bladder adhesion, RGT excision during CS
Czuczwar [24], 2016	NM/1	TVUS/6/None	Yes/Yes	US-guided intervention (KCl injection)/7	None	None	CS at 37 GW, live male, 3060 g
Lincenberg [7], 2016	NM/3	TVUS/10^+2^/AP, intraperitoneal hemorrhage	Yes/Yes	Surgical intervention (laparoscopy, laparotomy)/10^+2^	Persistent	Uterine rupture	CS at 23^+1^ GW, live female, 423 g
Vetter [26], 2016	Spontaneous/1	TVUS/5/VB	Yes/Yes (too early)	Surgical intervention (laparotomy)/NM	None	None	CS at 37^+1^ GW, live female, 3479 g
Miyague [25], 2018	NM/1	TVUS, MRI/6/None	Yes/Yes	US-guided intervention (combined KCL injection + EA)/NM	Growth with vascularity	RGT growth and AVM and MAP formation	Hysterectomy
Vikhareva [27], 2018	NM/1	TVUS/13/None	None/Yes	Expectant	Disappeared at 18 GW	None	VD at 39 GW, live male, 2985 g
Tymon-Rosario [28], 2018	NM/2	TVUS/NM/None	Yes/Yes	US-guided intervention (KCL injection)/10^+6^	N/M	Septic shock	Hysterectomy after UAE, D&C
Ashwini J Authreya [31], 2021	ART/1	TVUS/7^+6^/None	Yes/Yes	US-guided intervention (KCL injection)/NM	None	None	CS at 38 GW, a term healthy baby
Zheng-Yun Chen [29], 2021	Spontaneous/1	TVUS/8/None	Yes/Yes	Hyperosmolar glucose injection and EA/8^+2^, transcervical D&C	Disappeared at 20 GW	Vaginal bleeding	CS at 34^+2^ GW, a healthy baby PROM
Ouyang [30], 2021	ART/1	TVUS/6/None	Yes/Yes	Abortion (D&C)			D&C + UAE
Ouyang [30], 2021	ART/1	TVUS/6/VB	Yes/Yes	US-guided intervention (KCL injection)/NM	Persistent	Vaginal bleeding	IUP miscarriage at 14 GW
Ouyang [30], 2021	ART/1	TVUS/6^+2^/VB	Yes/Yes	Expectant/8	Persistent	Hysteroscopic excision of the CSP due to placenta accreta at 8 GW	
Ouyang [30], 2021	ART/1	TVUS/6/None	Yes/None	HIFU/7	Persistent	Hysteroscopic removal of RGT	Miscarriage of IUP at 7 GW
Ouyang [30], 2021	ART/1	TVUS/5^+5^/VB	Yes/Yes	D&C/13	NM	IUP and CSP miscarriage at 13 GW	
Ouyang [30], 2021	ART/1	TVUS/6/VB	Yes/Yes	Expectant	Disappeared at 20 GW	CSP miscarriage at 13 GW	CS at 29 GW, live female, 1300 g
Ouyang [30], 2021	ART/1	TVUS/6/VB	Yes/Yes	Expectant	NM	None	CS at 40 GW, live two females, 2900 g and 2200 g
Ouyang [30], 2021	ART/1	TVUS/5^+5^/VB	Yes/Yes	Expectant	NM	IUP miscarriage at 20 GW	CS at 36 GW, live female (CSP), 3000 g
Ouyang [30], 2021	ART/1	TVUS/6/VB	Yes/Yes	Expectant	NM	IIOC	Induced abortion at 22 GW
Ouyang [30], 2021	ART/2	TVUS/6/None	Yes/Yes	Expectant	Persistent	CSP miscarriage at 10 GW	CS at 37 GW, live male, 2600 g
Ouyang [30], 2021	ART/1	TVUS/6/None	None/Yes	Expectant	Disappeared at 22 GW	PROM	CS at 36 GW, live female, 2900 g
Ouyang [30], 2021	ART/1	TVUS/6^+5^/VB,AP	None/Yes	Expectant	Persistent	None	CS at 39 GW, live female 3900 g
Ouyang [30], 2021	ART/1	TVUS/6/VB	None/Yes	Expectant	Persistent	Placental abruption	CS at 24 GW
Ouyang [30], 2021	ART/4	TVUS/8^+5^/VB	None/Yes	Expectant	Disappeared at 16 GW	None	CS at 39 GW, live singleton, 2900 g
Ouyang [30], 2021	ART/1	TVUS/6^+2^/VB	None/Yes	Expectant	Persistent	Complete placenta previa	Emergency CS at 35 GW, live male, 2600 g
Ouyang [30], 2021	ART/1	TVUS/7^+1^/AP	None/Yes	Expectant	Persistent	PPROM	Induced abortion at 24 GW)
Ouyang [30], 2021	ART/1	TVUS/6/None	None/Yes	Expectant	Disappeared at 24 GW	None	CS at 39 GW, live male, 3150 g
Ouyang [30], 2021	ART/1	TVUS/5^+1^/VB	None/Yes	Abortion (D&C and UAE at 7 GW)	Removed		
Ouyang [30], 2021	ART/1	TVUS/4^+6^/AP	Yes/Yes	Expectant	NM	IUP miscarriage at 13 GW	Hysteroscopic removal of IUP
Ouyang [30], 2021	ART/1	TVUS/11/None	None/Yes	Expectant	None	Uterine rupture at 12 GW	Laparotomy repair
Laing-Aiken [32], 2020	Spontaneous/1	TVUS/9/VB	Yes/Yes	Surgical intervention (D&C, laparotomy)/9	Removed	PPROM	CS at 28^+1^ GW, live male, 1200 g, bilateral uterine artery ligation

AP—abdominal pain; ART—assisted reproduction techniques; AVM—arteriovenous malformation; CS—cesarean section; CSP—cesarean section pregnancy; D&C—dilation and curettage; EA—embryo aspiration; GW—gestational weeks; IIOC—incompetent internal os of cervix; HIFU—high-intensity focused ultrasound; IUP—intrauterine pregnancy; KCL—potassium chloride; MAP—morbidly adherent placenta; MRI—magnetic resonance imaging; NM—not mentioned; PROM—premature rupture of membrane; PPROM—preterm premature rupture of membrane; TVUS—transvaginal ultrasonography; UAE—uterine artery embolization; VB—vaginal bleeding.

## Data Availability

The datasets used in this study are available from the corresponding author.

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
