# Peer review of "Successful Full-Term Delivery via Selective Ectopic Embryo Reduction Accompanied by Uterine Cerclage in a Heterotopic Cesarean Scar Pregnancy: A Case Report and Literature Review"

_diagnostics, 2022, doi:10.3390/diagnostics12030762_

Round 1

Reviewer 1 Report

Dear Authors: Excellent presentation and decisions for this challenging case. You choose perfectly the words and the recommendation for this unusual case.

Best regards,

The reviewer

Author Response

Thank you for your compliment. Your comments encouraged us a lot.

Reviewer 2 Report

Dear Authors,

very interesting case, but I havr some comments:

  1. In Introduction you wrote that you performed embryo aspiration in case report case you wrote that you removed embryo, which version in correct?
  2. Which technique was used to place the cervical cerclage?
  3. In line 86 you wrote that patient was delivered at another hospital and in line 90 is written that patient did not require hospitalization...maybe I misunderstood something...
  4. In discussion lines 175-205 are the same as lines 130-160...why?

Author Response

Reviewer #2

  1. In Introduction you wrote that you performed embryo aspiration in case report case you wrote that you removed embryo, which version in correct?

<Response> We apologize for confusion in the meanings. As pointed out by the reviewer #2, we changed the word ‘removed’ into ‘aspirated’ at lines 55-56, page 2 in the revised manuscript as shown below:

“After cardiac arrest of the CSP, the expired embryo was completely aspirated without affecting the placenta around the CSP.”

  1. Which technique was used to place the cervical cerclage?

<Response> We followed McDonald operation for the cervical cerclage. Details were described at lines 75-79, page 3 in the revised manuscript as shown below:

“Double ligations of McDonald operation using braided polyester thread (EthibondTM) were implemented in the cervical cerclage [4]. A previous retrospective study reported that the braided thread suture in the cervical cerclage showed the improvement in neonatal survival, prevention of preterm birth before 28 GW, less PPROM, and maternal febrile morbidity, compared to Mersilene tape [5].”

<References>

[4] Berghella, V.; Szychowski, J.M.; Owen, J.; Hankins, G.; Iams, J.D.; Sheffield, J.S.; Perez-Delboy, A.; Wing, D.A.; Guzman, E.R.; Consortium, V.U.T. Suture type and ultrasound-indicated cerclage efficacy. J Matern Fetal Neonatal Med 2012, 25, 2287-2290, doi:10.3109/14767058.2012.688081.

[5] Leah Bernard, L.P., Vincenzo; Berghella, O.R., Suneeta Mittal, Sean Daly, M. Vaarasmaki, Amanda Cotter,; Ricardo Gomez, W.P., Surasith Chaithongwongwatthana, Jorge; Tolosa, E. Effect of suture material on outcome of cerclage in women with a dilated cervix in the 2nd trimester: Results from the expectant management compared to physical exam-indicated cerclage (EM-PEC) international cohort study. American Journal of Obstetrics and Gynecology 2006, 195, doi:DOI:https://doi.org/10.1016/j.ajog.2006.10.342.

  1. In line 86 you wrote that patient was delivered at another hospital and in line 90 is written that patient did not require hospitalization...maybe I misunderstood something...

<Response> We appreciate the reviewer’s comment. The reason why we wrote ‘hospitalization’ in the manuscript was that there was no need of longer hospitalization in our case. The patient had been hospitalized for only four days after cesarean delivery.

For clarification, we revised the paragraph at lines 97-100, page 4 as described below:

“After delivery of a baby, massive bleeding was developed at the site of CSP. The RGT was removed and bleeding focus was controlled by multiple sutures. Her vital sign was stable with red blood cells transfusion. She did not need further intensive care. The RGT was histologically confirmed as an AVM.”

  1. In discussion lines 175-205 are the same as lines 130-160...why?

<Response> We apologize for our mistake during preparation of the manuscript. The indicated paragraph was entirely removed in the revised manuscript.

Additional amendments

  1. (page 4; line 93) We revised delivery gestational weeks and neonatal bodyweight in more detail as below:

Delivery was performed at another hospital for private reasons. A healthy female baby weighing 2,415 g was delivered via elective cesarean section at 37+6 GW.

  1. We appended an acknowledgement to present appreciation to the contributor who had successful completed the delivery of our patient.

Acknowledgments

We appreciate Prof. Jong Kwan Jun (the Seoul National University Hospital, South Korea) for helpful advice during preparation of the manuscript. And he had successfully completed the delivery of our patient.

  1. We also applied some minor revisions colored with yellow